# RETHINKING SKIP CONNECTION MODEL AS A LEARNABLE MARKOV CHAIN

**Dengsheng Chen**[*]
Meituan
chendengsheng@meituan.com

**Jie Hu**[*]
State Key Laboratory of Computer Science, ISCAS
University of Chinese Academy of Sciences
hujie@ios.ac.cn

**Wenwen Qiang**
University of Chinese Academy of Sciences
Institute of Software Chinese Academy of Sciences
wenwen2018@iscas.ac.cn

**Xiaoming Wei**
Meituan
weixiaoming@meituan.com

**Enhua Wu**[†]
State Key Laboratory of Computer Science, ISCAS
University of Chinese Academy of Sciences
University of Macau
ehwu@um.edu.mo

## ABSTRACT

Over the past few years afterward the birth of ResNet, skip connection has become the defacto standard for the design of modern architectures due to its widespread adoption, easy optimization, and proven performance. Prior work has explained the effectiveness of the skip connection mechanism from different perspectives. In this work, we deep dive into the model's behaviors with skip connections which can be formulated as a learnable Markov chain. An efficient Markov chain is preferred as it always maps the input data to the target domain in a better way. However, while a model is explained as a Markov chain, it is not guaranteed to be optimized following an efficient Markov chain by existing SGD-based optimizers prone to getting trapped in local optimal points. In order to move towards a more efficient Markov chain, we propose a simple routine of penal connection to make any residual-like model become a learnable Markov chain. Aside from that, the penal connection can also be viewed as a particular model regularization and can be easily implemented with one line of code in the most popular deep learning frameworks. The encouraging experimental results in multi-modal translation and image recognition empirically confirm our conjecture of the learnable Markov chain view and demonstrate the superiority of the proposed penal connection.

## 1 INTRODUCTION

Over the last decade, deep learning has been dominant in many tasks, including image recognition (Voulodimos *et al.*, 2018), machine translation (Singh *et al.*, 2017), speech recognition (Zhang *et al.*, 2018), etc. Many SGD-based methods and excellent network structures come to the fore (Alom *et al.*, 2019). Among them, skip connection (He *et al.*, 2016) is a widely-used technique to improve the performance and the convergence of deep neural networks. Aided by the skip connection, models with very deep layers can be easily optimized by SGD-based methods (Amari, 1993), e.g., vanilla SGD (Cherry *et al.*, 1998), Momentum SGD (Sutskever *et al.*, 2013), Adagrad (Lydia and Francis, 2019), Adam (Kingma and Ba, 2014).

Recently, many theoretical explanations of how it works have been largely underexplored (Li and Yuan, 2017; Allen-Zhu *et al.*, 2019). In this work, we continued to explore the behaviors of the model with skip connection and view it as a **learnable Markov chain** (short for Markov

---

[*]Equal contribution.
[†]Corresponding author. The work is supported in part by NSFC Grants (62072449).

chain) (Gagniuc, 2017) . To our best knowledge, it is the first time to analyze from this perspective. In the conception of the Markov chain, the output of a residual block is noted as the *predicted direction* with respect to the input. For better elaboration, we introduce another term *ideal direction*. The ideal direction always points to a more accurate direction than the predicted direction, which can translate an input to the target domain in a more efficient way. Then, we define an indicator $\varepsilon$ to reflect how efficient a learned Markov chain is, based on the angle between the predicted and ideal direction. In contrast to the original predicted direction, an efficient Markov chain with an ideal direction is preferred since it always maps the input to the target domain in a better way. However, we are aware that existing SGD-based optimizers are quite lazy to update the model following an efficient Markov chain, which hinders the upper bound performance.

To train a more efficient Markov chain, we propose a very simple routine of penal connection to convert a residual-like model to a Markov chain by just adding one line of code in existing deep learning frameworks. On the one hand, the penal connection is capable of enforcing the optimizer to update the model following the rules of the efficient Markov chain. On the other hand, it can be viewed as a type of additional model regularization, which alleviates over-fitting and enhances generalization. Compared with the original residual-like model, the Markov chain also has more benefits in deeper networks that suffer from performance degradation corresponding to more learnable parameters. The experimental results in multi-modal translation and image recognition not only demonstrate the feasible analysis of regarding a residual-like model as a Markov chain but also examine the superiority of the proposed penal connection throughout the optimization process.

Our main contributions can be summarized in two folds. First, we present a new perspective to understand the skip connection model as a learnable Markov chain and carry out exhaustive theory analysis and experimental verification. Second, we propose the penal connection, a simple method to enable a network to be optimized within a more efficient Markov chain, which can substantially improve performances both in the fields of NLP and CV.

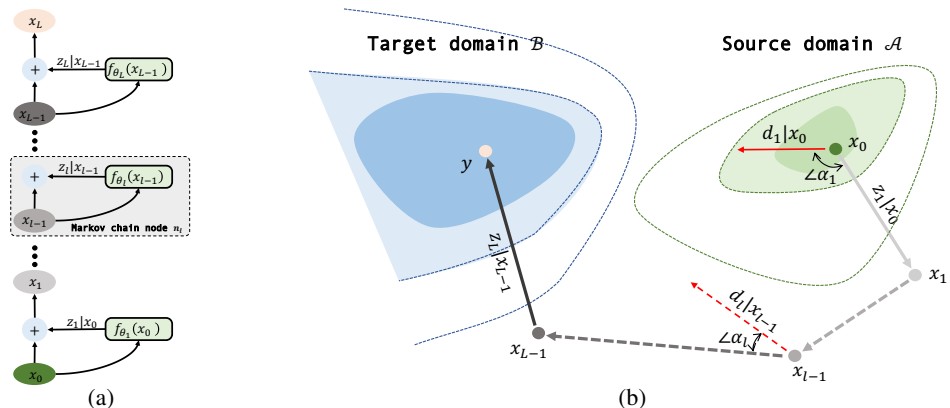

Figure 1: A model $\mathcal{M}$ with $L$ skip connections can be recognized as a Markov chain $\mathcal{C}$ consisting of $L$ nodes. The forward pass is corresponding to a Markov process. As shown in Fig. 1(a), a skip connection along with a residual-like block $f_{\theta_l}(\cdot)$ builds up a Markov chain node $n_l$ (the gray dash box in middle). The input of $n_l$ is $x_{l-1}$, i.e., the output of the previous Markov node. The output of $n_l$ can be formulated by $x_l = x_{l-1} + z_l|x_{l-1}$, where $z_l|x_{l-1} = f_{\theta_l}(x_{l-1})$ is the predicted direction by residual-like block. As shown in Fig. 1(b), guided by $z_{l,l\in 1,\cdots,L}$, the input data $x_0 \in \mathcal{A}$ can gradually shift to the target label $y \in \mathcal{B}$ along the learned Markov chain. The red dashed arrow $d_l|x_{l-1}$ is the ideal direction with respect to $x_{l-1}$ and $z_l$.

## 2 METHOD

In this section, we first reformulate the residual-like model as a Markov chain and introduce $\varepsilon$ to reflect the efficiency of a learned Markov chain. Then, we also define a $\delta$-convex chain and make the convergence proof based on it. Before exploring the optimization algorithm, the dilemma between the behavior of an efficient Markov chain and existing backward-propagation algorithms is thoroughly discussed. Lastly, we propose a penal connection mechanism to boost the performance of the Markov chain.

## 2.1 THE LEARNABLE MARKOV CHAIN

Similar to the definition of a traditional Markov chain in Gagniuc (2017), a learnable Markov chain can be defined as:

**Definition 1** (*The learnable Markov chain:* $\mathcal{C}_L$). *A learnable Markov chain $\mathcal{C}_L$ is a stochastic model with learnable parameters $\{\theta_1, \cdots, \theta_L\}$ describing a sequence of L possible events in which the state $x_l$ of the current event only depends on the state $x_{l-1}$ attained in the previous event.*

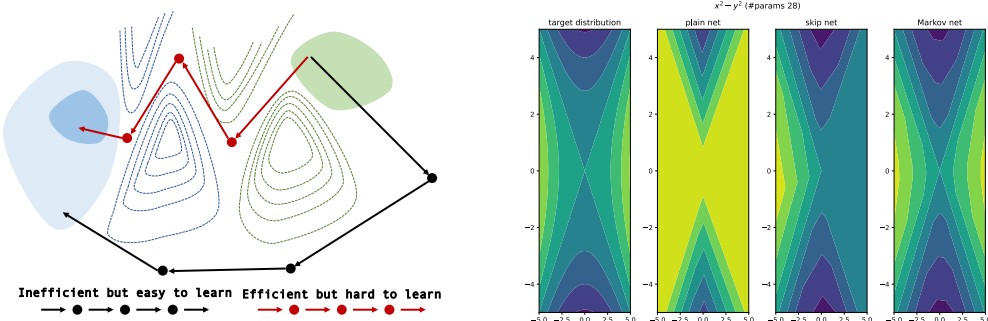

(a) The influence of efficient/inefficient Markov chain. (b) A toy model to demonstrate the importance of a suitable $\varepsilon$.

Figure 2: A simple task to show the performance difference between plain net, skip net, and our proposed Markov net. In this task, we build a model which consists of three fully connected layers with 28 learnable parameters to shift coordinate $(x, y)$ to a target distribution, which can be formulated as $x^2 - y^2$. More details are listed in supplementary materials. We update the model with SGD optimizer for 10000 steps. As shown in Fig.2(b), Markov net can better complete this task.

As shown in Fig. 1(a), $\mathcal{M}_L$ indicates a model with $L$ residual-like blocks, and can also be considered as a learnable Markov chain $\mathcal{C}_L$ with $L$ nodes since the output state (i.e. output feature map of a residual block) $x_l$ of node $n_l$ only depends on the state $x_{l-1}$ from the previous node $n_{l-1}$. As a result, a forward pass through $\mathcal{M}_L$ can be viewed as a Markov process, as shown in Fig. 1(b). In a bit more detail, the corresponding Markov chain with respect to the input data $x_0 \in \mathcal{A}$ is formulated as:

$$\mathcal{C}_L(x_0 \to x_L) := x_0 \xrightarrow{+z_1|x_0} x_1 \to \cdots x_l \xrightarrow{+z_l|x_{l-1}} x_{l+1} \to \cdots x_{L-1} \xrightarrow{+z_L|x_{L-1}} x_L \quad (1)$$

where $f_{\theta_l}$ is a feature transformation function by $i$-th residual-like block in $\mathcal{M}_L$, and is also the $i$-th chain node in $\mathcal{C}_L$ with learnable parameters $\theta_l$. $z_l|x_{l-1} = f_{\theta_l}(x_{l-1})$ is the predicted direction for previous state $x_{l-1}$ in node $n_l$. Correspondingly, the ideal direction $d_l|x_{l-1}$ with respect to $x_{l-1}$ and $z_l$ can be defined as:

**Definition 2** (*The ideal direction:* $d_l|x_{l-1}$). *Assume the function $\ell$ measures the distance between two variables, and if $\ell(a, c) \geq \ell(b, c)$, then $\ell(a, c) \geq \ell(\mu a + (1-\mu)b, c) \geq \ell(b, c)$, where $\mu \in [0, 1]$. $d_l|x_{l-1}$ is an ideal direction with respect to $x_{l-1}$ and $z_l$ as long as:*

$$\ell(x_{l-1} + \eta d_l|x_{l-1}, y) \leq \ell(x_{l-1} + \eta z_l|x_{l-1}, y). \quad (2)$$

*where $\eta$ is a small step size.*

Obviously, given $x_{l-1}$, $z_l$ and $\ell$, the ideal direction $d_l$ is still not unique because anyone who can outperform the predicted direction $z_l$ under the measurement of $\ell$ is qualified, as Eq. 2 holds. We collect all the ideal directions for $x_{l-1}, z_l$ under $\ell$ as $\mathcal{D}_{\ell, z_l|x_{l-1}}$.

**Lemma 1** *If $d_l \in \mathcal{D}_{\ell, z_l|x_{l-1}}$, then $d'_l = \mu d_l + (1-\mu)z_l \in \mathcal{D}_{\ell, z_l|x_{l-1}}$, where $\mu \geq 0$.*

**Proof 2.1** *Since $d_l$ is an ideal direction, then:*

$$\ell(x_{l-1} + \eta d_l, y) \leq \ell(\mu(x_{l-1} + \eta d_l) + (1 - \mu)(x_{l-1} + \eta z_l), y)$$
$$= \ell(x_{l-1} + \eta(\underbrace{\mu d_l + (1 - \mu)z_l}_{d_l'}), y)$$
$$\leq \ell(x_{l-1} + \eta z_l, y), \tag{3}$$

*thus, $d_l' = \mu d_l + (1 - \mu)z_l$ is an ideal direction with respect to $z_l$ and $x_{l-1}$.*

Different function $f_{\theta_l}$ takes discrepant effects on the set $\mathcal{D}_{\ell, z_l | x_{l-1}}$. $f_{\theta_l}$ is used to be a sequence of convolutions (e.g. residual block in ResNet (He *et al.*, 2016))or a popular *Transformer* introduced in Vaswani *et al.* (2017), as long as it can make $\mathcal{C}_L$ shift input $x_0 \in \mathcal{A}$ to the target domain $\mathcal{B}$ correctly. Importantly, we next devise an intuitive indicator $\varepsilon$ to reflect how the efficiency of $\mathcal{C}_L(x_0 \to x_L)$:

**Definition 3** *(**The efficiency of Markov chain:** $\mathcal{C}_L(x_0 \to x_l)$). The efficiency indicator $\varepsilon$ measures an average cosine similarity between $z_l | x_{l-1}$ and $d_l | x_{l-1}$:*

$$\varepsilon := \frac{1}{L} \sum_{l=1}^{L} \cos \alpha_l = \frac{1}{L} \sum_{l=1}^{L} \langle \vec{z_l}, \vec{d_l} \rangle. \tag{4}$$

*where $\vec{v}$ is the normalized tensor of $v$, defined as $\vec{v} := v/\|v\|_2$. $\angle \alpha_l$ is the angle between $z_l | x_{l-1}$ and $d_l | x_{l-1}$.*

From the definition, a larger $\varepsilon$ indicates a smaller $\angle \alpha_l$ between the predicted direction $z_l$ and the ideal direction $d_l$, mirroring a more efficient Markov chain and vice versa. More specifically, if $z_l$ always has a positive fraction on $d_l$ for all nodes, we call this is a convex chain which formally defined as follows:

**Definition 4** *($\delta$-**convex chain**). If $\exists \delta > 0$, $\forall n_l \in \mathcal{C}_L(x_0 \to x_L)$, $\langle z_l, d_l \rangle > \delta \|d_l\|_2^2$, then $\mathcal{C}(x_0 \to x_L)$ is dubbed as a $\delta$-convex chain.*

**Lemma 2** *If $\mathcal{C}_L(x_0 \to x_L)$ is $\delta$-convex, $\varepsilon > 0$.*

**Proof 2.2** *Since $\mathcal{C}_L(x_0 \to x_L)$ is $\delta$-convex, then for all $l$:*

$$\langle z_l, d_l \rangle = \langle \|z_l\|_2 \vec{z_l}, \|d_l\|_2 \vec{d_l} \rangle = \|z_l\|_2 \|d_l\|_2 \langle \vec{z_l}, \vec{d_l} \rangle > \delta \|d_l\|_2^2 > 0 \tag{5}$$

*It is noteworthy that $\|z_l\|_2 \|d_l\|_2 > 0$, thus $\forall n_l$, $\langle \vec{z_l}, \vec{d_l} \rangle > 0$, which yields $\varepsilon = \frac{1}{L} \sum_{l=1}^{L} \langle \vec{z_l}, \vec{d_l} \rangle > 0$.*

Lemma 2 tells that if $\mathcal{C}_L(x_0 \to x_L)$ is $\delta$-convex, it must be an efficient Markov chain, while the reverse is not necessarily true. As shown in Fig. 1(b), the plotted chain is an efficient Markov chain, but it is not a $\delta$-convex chain since there does not exist a non-negative $\delta$ making $\angle \alpha_1$ satisfy Definition 4.

If $\mathcal{C}_L(x_0 \to x_L)$ is $\delta$-convex, then in every node $n_l$, a positive fraction of the predicted direction $z_l$, i.e., $z_l \cos \angle \alpha_l$, is pointing to the same direction as $d_l$. Given an appropriate step size $\eta$, the input could eventually arrive at the target domain, despite along with a winding path. Fig. 2(a) also illustrates a simple model to verify it. The source input gets closer to target domain $\mathcal{B}$ in every step while moving along a $\delta$-convex Markov chain (visualization in brown).

Formally, we have concluded the following lemma for ensuring convergence.

**Lemma 3** *For each chain node $n_l \in \mathcal{C}_L(x_0 \to x_L)$, consider the forward process $x_l = x_{l-1} + z_l$, where $\mathbb{E}[z_l] = d_l, \mathbb{E}[\|z_l\|_F^2] \leq Z^2$. Suppose for all nodes, $x_l$ is always inside the $\delta$-convex region with diameter $D$, i.e., $\|x_t - y\|_F \leq D$. Then, for any $a > 0$ and any $L$ such that $L^a \log L \geq \frac{D^2 \delta^2}{(1+a)Z^2}$, we have $\mathbb{E}[\|x_L - y\|_F^2] \leq \frac{(1+a) \log L Z^2}{\delta^2 L}$.*

The proof of Lemma 3 appears in Appendix B. Notably, Lemma 3 does not imply that $x_L$ equals to $y$. It only describes that $x_L$ is sufficiently close to $y$ if $z_l$ can predict the correct direction for $x_{l-1}$. For a longer chain (i.e. deeper network with larger $L$), the $x_L$ will be restricted closer to target $y$ with a relatively less error. Taken together, Lemma. 3 guarantees that a Markov chain $\mathcal{C}_L$ can shift source input $x_0 \in \mathcal{A}$ to target domain $\mathcal{B}$ through $L$ nodes in an efficient way if $\mathcal{C}_L$ is $\delta$-convex. Until now, we can trustingly settle down to optimize $\mathcal{M}_L$ from the conception of $\mathcal{C}_L$.

## 2.2 MARKOV CHAIN OPTIMIZATION

Before exploring the optimization method of a Markov chain, it is necessary to have a careful discussion about the efficiency of a Markov chain taking effect in the optimization process. According to aforesaid Lemma. 3, if all parameters $\theta_l$ are optimized in a way that makes $\mathcal{C}_L(x_0 \rightarrow x_L)$ always be a $\delta$-convex chain for any input $x_0$, the convergence speed for optimization appears to be substantially improved, and the final performance as well. However, we caution that a $\delta$-convex chain is not guaranteed by existing SGD-based optimizers, such as SGD and Adam. Actually, the existing optimizers prefer towards to an efficient or even an inefficient Markov chain, where the land scope of the loss is more smooth, instead of a $\delta$-convex one. After discovery and practice, we find SGD-based optimizers have the potential to bridge the gap which will be discussed later. The diagrammatic sketch of efficient and inefficient Markov chain is illustrated in Fig. 2(a), which can better help to understand.

Intuitively, while $\varepsilon \rightarrow 1$, an efficient Markov chain acts more like a $\delta$-convex chain and can obtain its good properties. However, a too large $\varepsilon$ is a disaster for existing optimizers, resulting in hard convergence. Hence, the main idea to optimize a Markov chain is to train a "reasonable" efficient Markov chain, i.e. let $\|\varepsilon\|_2^2$ no larger than a given threshold $\rho$. In this way, the objective function can be formulated as:

$$\mathcal{L} = \mathcal{L}_{\mathcal{M}_L}(x_L, y) + \lambda \|\varepsilon\|_2^2. \tag{6}$$

The first item of Eq. 6 is the original loss function of the model $\mathcal{M}_L$ and the second item is the additional penalization with $\lambda$ to hold $\|\varepsilon\|_2^2 \leq \rho$. If set $\lambda = 0$ in Eq. 6, the objective function degenerates into a plain mode without any remedy toward forming a more efficient Markov chain. Instead, if set $\lambda > 0$, the penalization will take effect, suggesting enable to obtain a more efficient Markov model. The toy example in Fig. 2(b) visualizes the significant effect of this penalization term over the plain net (MLPs) and skip net (MLPs with skip connection).

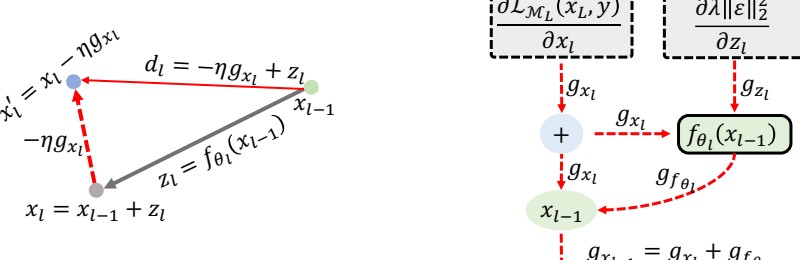

(a) The proposed ideal direction.

(b) The backward propagation of gradient within a Markov chain node.

Figure 3: A visualization of ideal direction computed based on $g_{x_l}$ in Fig. 3(a). It is worth noting that $x_l'$ can be viewed that we update $x_l$ by $g_{x_l}$ with a small learning rate $\eta$. The backward propagation of gradients within a Markov chain node is plotted in Fig. 3(b). Compared with the residual-like model, we only add an additional gradient $g_{z_l}$ while computing the gradient with respect to $f_{\theta_l}$.

In order to solve Eq. 6, the correct ideal direction $d_l$ with respect to $x_{l-1}$ is required to figure out. As discussed previously, $d_l$ is not unique, thus the different definition of $\ell$ leads to a different set of $d_l$. Actually, since the feature space of $x_l$ always lies in very high dimension space, it would be a great challenge to find a suitable $\ell$ to define an ideal direction $d_l$.

As for a chain node $n_l$, we reuse the target loss function $\mathcal{L}_{\mathcal{M}_L}$ to build a valid $\ell$ function:

$$\ell(x_l, y) := \mathcal{L}_{\mathcal{M}_L}(\mathcal{C}_L(x_l \rightarrow x_L), y). \tag{7}$$

$x_l$ is forward along the rest chain nodes $n_{l+1}, \cdots, n_L$, and the final output $x_L$ is taken to compute the loss between $y$ by $\mathcal{L}_{\mathcal{M}_L}$. This way, the gradient with respect to $x_l$ is:

$$g_{x_l} := \frac{\partial \ell(x_l, y)}{\partial x_l} = \frac{\partial \mathcal{L}_{\mathcal{M}_L}(x_L, y)}{\partial x_l} \tag{8}$$

Hence, $g_{x_l}$ can be obtained while the backward propagation during the training process and an ideal direction $d'_l$ based on $g_{x_l}$ is $z_l - \eta g_{x_l}$, where $\eta$ is a small step size, as shown in Fig. 3(a).

**Proof 2.3** *Since $x_l = x_{l-1} + z_l$ and $\ell(x_l - \eta g_{x_l}, y) < \ell(x_l, y)$ always holds, set $d'_l := z_l - \eta g_{x_l}$, then*

$$\ell(x_{l-1} + d'_l, y) = \ell(x_{l-1} + z_l - \eta g_{x_l}, y) = \ell(x_l - \eta g_{x_l}, y) < \ell(x_l, y) = \ell(x_{l-1} + z_l, y) \tag{9}$$

*Thus, $d'_l = z_l - \eta g_{x_l}$ is an ideal direction for $x_{l-1}$.*

From Lemma 1, we known that $d_l = \mu d'_l + (1 - \mu)z_l = z_l - \eta\mu g_{x_l}$ is also an ideal direction, i.e., $z_l - \eta\mu g_{x_l} \in \mathcal{D}_{\ell, z_l | x_{l-1}}$. Assume $\|g_{x_l}\|_2 \geq 1$, then we can set $\mu = \frac{1}{\|g_{x_l}\|_2} \in [0, 1]$, yields $d_l = z_l - \eta\vec{g}_{x_l}$. Since $\eta$ is small, $\vec{d_l}$ can be approximately expressed as $\vec{z_l} - \eta\vec{g}_{x_l}$, and $\varepsilon$ can be reformulated as:

$$\varepsilon = \frac{1}{L}\sum_{l=1}^{L}\langle \vec{z_l}, \vec{d_l} \rangle \approx \frac{1}{L}\sum_{l=1}^{L}\langle \vec{z_l}, \vec{z_l} - \eta\vec{g}_{x_l} \rangle = \frac{1}{L}\sum_{l=1}^{L}\langle \vec{z_l}, \vec{z_l} \rangle + \langle \vec{z_l}, -\eta\vec{g}_{x_l} \rangle$$

$$= \frac{1}{L}\sum_{l=1}^{L} 1 + \langle \vec{z_l}, -\eta\vec{g}_{x_l} \rangle = 1 + \eta\underbrace{\frac{1}{L}\sum_{l=1}^{L}\langle \vec{z_l}, -\vec{g}_{x_l} \rangle}_{\varepsilon'} \tag{10}$$

where $\varepsilon'$ will be used in the following experiments instead of $\varepsilon$ as it indicates the same efficiency properties of the Markov chain but is easier to understand (a larger $\varepsilon'$ indicates a more efficient $\mathcal{C}_L$).

Then, the gradient to $z_l$ can be computed as:

$$g_{z_l} := \frac{\partial \mathcal{L}_{\mathcal{M}_L}(x_L, y)}{\partial x_l}\frac{\partial x_l}{z_l} + \frac{\partial \lambda \|\epsilon\|_2^2}{\partial z_l}$$

$$= g_{x_l}\frac{\partial(x_{l-1} + z_l)}{\partial z_l} + \lambda\frac{\partial \|\sum_{l=1}^{L} 1 + \eta\langle \vec{z_l}, -\vec{g}_{x_l} \rangle\|_2^2}{\partial z_l}$$

$$\approx g_{x_l} + \lambda\frac{\partial \sum_{l=1}^{L} \|1 + \eta\langle cz_l, -\vec{g}_{x_l} \rangle\|_2^2}{\partial z_l} \tag{11}$$

$$= g_{x_l} + \lambda\frac{\partial \|1 + \eta\langle cz_l, -\vec{g}_{x_l} \rangle\|_2^2}{\partial z_l}$$

$$= g_{x_l} + 2\lambda(-\eta\vec{g}_{x_l})(1 + \langle cz_l, -\eta\vec{g}_{x_l} \rangle)$$

$$= g_{x_l} + 2\lambda(-\eta\vec{g}_{x_l}) + 2\lambda c\eta^2 z_l$$

$$= (1 - \frac{2\lambda\eta}{\|g_{x_l}\|_2})g_{x_l} + 2\lambda c\eta^2 z_l$$

$$\approx g_{x_l} + \tau z_l \tag{12}$$

where $\tau$ is a hyper-parameter and $c = \frac{1}{\|z_l\|_2}$ in Eq. 11 can be regarded as a constant value to simplify the gradient derivation process. The analysis of this estimation can be found in Appendix C. Despite lots of hyper-parameters introduced for facilitating derivation, e.g., $\lambda, \eta, c, \rho$, we only need to specify a single hyper-parameter $\tau$ in the final formulation, which relieves the heavy burden from a hyper-parameter sweep. We dubbed this special optimization method as penal connection, which seems to add a simple penalization on the norm of $z_l$ as a type of model regularization. The compute graph has been plotted in Fig. 3(b), and it can be easily implemented based on the PyTorch framework by adding one line of code (see Algo. 1[1]).

---

[1] Tips: $z_l$.register_hook(lambda $g_{x_l}$: $g_{x_l} + \tau z_l$) is not allowed which could lead to a memory leak.

---

**Algorithm 1** Pseudo code of penal connection in a PyTorch-like style.

---

$z_l = f_{\theta_l}(x_{l-1})$
\# Only following line is added to register a hook
$z_l$.register_hook(lambda $g_{x_l}$, $z_l=z_l$.detach().clone(): $g_{x_l} + \tau z_l$)
$x_l = x_{l-1} + z_l$

---

Lastly, due to the various choice of $\ell$, the gradient to $z_l$ is correspondingly discrepant. Further community exploration about more reasonable and effective ways to compute the ideal direction is of great value so continually push forward the performance of the Markov chain.

## 3 EXPERIMENTS

In this section, we conduct intensive experiments to demonstrate the superiority of the Markov chain in the field of *Natural Language Processing* and *Computer Vision*. Unless otherwise specified, all the residual-like blocks in $\mathcal{M}$ are converted to a Markov chain node $n_l$ in $\mathcal{C}$.

Specifically, in transformer block (Vaswani *et al.*, 2017) which consists of a multi-head self-attention module (MSA) and a feed-forward network (FFN), we convert it as two chain nodes as both MSA and FFN employ skip connection respectively.

For simplicity, we set $\tau$ the same for all nodes in a Markov chain. All experiments are carried out by publicly available projects implemented by PyTorch (Paszke *et al.*, 2017) on a device equipped with 8 NVIDIA-A100 GPUs. More experimental details can be found in the supplementary materials.

### 3.1 MULTI-MODAL TRANSLATION

Multi-modal translation requires a model to be capable of translating the text information from the source language $\mathcal{A}$ to the target language $\mathcal{B}$. How the efficient/inefficient Markov chain performs during the training process can be clearly observed in this experiment.

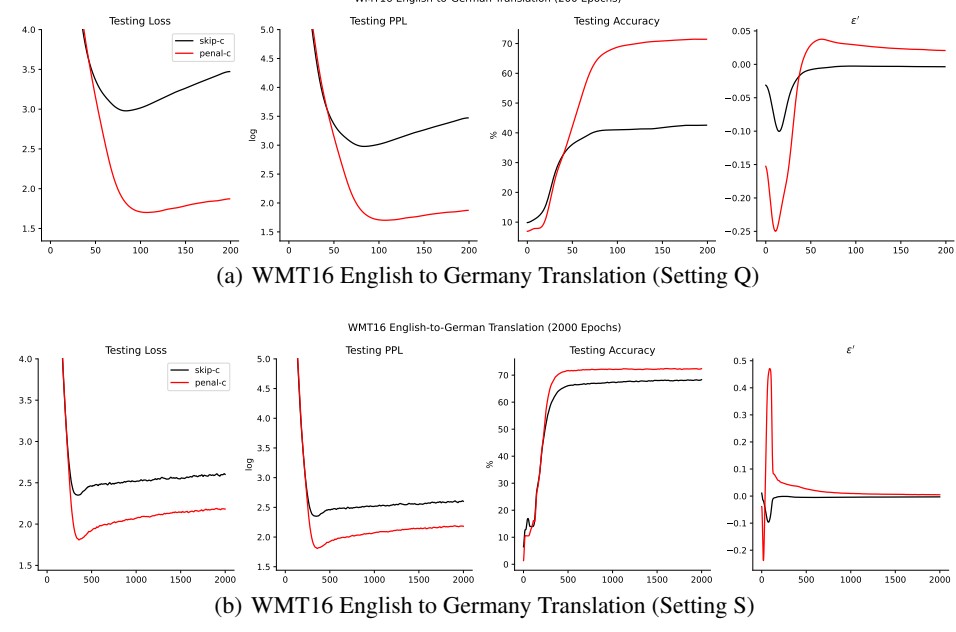

Figure 4: The testing curve across different epochs on WMT16 English to Germany translation tasks (Germany to translation curve can be found in supplementary materials).

**Dataset.** WMT16 (Bojar *et al.*, 2016) is a widely used translation dataset based on the data from *statmt.org*, which contains various interesting translation tasks on specified domains. Here, we are focusing on the news translation tasks between English and Germany. The text for all the test sets is drawn from news articles.

| Task | Settings | | PPL | | | Accuracy(%) | | |
|---|---|---|---|---|---|---|---|---|
| | WarmUp | Total | Trans. | Markov. | Improv. | Trans. | Markov. | Improv. |
| WMT-16 DE/EN (Q) | 35 | 200 | 33.3 | 7.9 | ↓ **25.4** | 36.4 | 70.0 | ↑ **33.6** |
| WMT-16 DE/EN (S) | 350 | 2000 | 17.3 | 11.3 | ↓ **6.0** | 66.4 | 70.4 | ↑ **4.0** |
| WMT-16 EN/DE (Q) | 35 | 200 | 31.8 | 6.3 | ↓ **25.5** | 43.1 | 71.8 | ↑ **28.7** |
| WMT-16 EN/DE (S) | 350 | 2000 | 13.4 | 8.5 | ↓ **4.9** | 68.7 | 72.8 | ↑ **4.1** |

Table 1: Testing PPL (the lower the better) and accuracy (the higher the better) of Transformer (Trans.) and counterpart Markov chain (Markov.).

| Model | Baseline | $3 \times 10^{-9}$ | $3 \times 10^{-8}$ | $3 \times 10^{-7}$ | $3 \times 10^{-6}$ | Improv. |
|---|---|---|---|---|---|---|
| ResNet50 | 79.2 | 79.0 | 79.4 | 79.2 | 79.3 | **+0.2** |
| ViT-S/16-224 | 77.8 | 78.0 | 78.1 | 78.0 | 77.8 | **+0.3** |
| DeIT-S/16-224 | 79.9 | 79.9 | 80.1 | 80.0 | 79.8 | **+0.2** |
| Swin-S/4-7-224 | 83.3 | 83.2 | 83.5 | 83.3 | 83.1 | **+0.2** |

Table 2: The top-1 accuracy on ImageNet-1k on different architectures with different $\tau$.

**Implementation details.** We adopt the most widely used benchmark *Transformer* (Vaswani *et al.*, 2017) as our strong baseline. The embedding size $d_{emb}$ is 512, and the source and target embedding layers are shared. We empirically set $\tau$ to $3 \times 10^{-4}$, which generalizes well to all translation tasks. Here, we opt for the mutual translation tasks between English and Germany for validation. All the models are trained using Adam optimizer with $\beta_1 = 0.9, \beta_2 = 0.98$. We use a batch size of 1024 and weight decay of 0.05 and other recipes for training are identical to the original implementation (Vaswani *et al.*, 2017). We set up two regular training settings, Q and S separately (see Table 1).

**Result analysis.** From Fig. 4 and Tab. 1, a few patterns can be observed. One is that the Markov chain with penal connection converges faster than the residual counterpart. Surprisingly, the Markov chain training with 200 epochs can outperform the baseline model well-training with 2000 epochs (10× training schedule), demonstrating the good merits of the proposed method. It is worth noting that the transformer model does not saturate without adequate training schedule length. The $\varepsilon'$ value plotted in Fig. 4 may illustrate this phenomenon. From the curve, we find that the transformer model is more likely to be an inefficient Markov chain whose $\varepsilon'$ is always negative. Aided by the proposed penal connection, it becomes an efficient Markov chain whose $\varepsilon'$ is mostly positive. Furthermore, we observed that at the early training stage, $\mathcal{C}_L$ is an inefficient chain, which is even worse than the baseline. After several epochs, it conversely turns to become a $\delta$-convex chain (with a large $\varepsilon'$). After that, $\mathcal{C}_L$ decays to a less efficient Markov chain with a small positive $\varepsilon'$. This phenomenon also confirms our analysis in the method section: Moving along with a $\delta$-convex chain can help converge to the target in the most efficient way. However, it may not be the best way under the existing optimizer. In order to achieve higher performance, the model will be pushed to be a less efficient chain. As for the baseline model, we find that it always acts as an inefficient Markov chain which more easily gets trapped in local minima.

## 3.2 IMAGE CLASSIFICATION

Different from translation tasks, the gradient for image classification is quite sparse and the redundant parameters often guarantee the optimizer towards to an effective Markov chain, even to a $\delta$-convex chain, suggesting that even without the penal connection, the model can also be optimized efficiently. Despite that, we still observe a slight $\tau$ (smaller than $10^{-7}$) can further improve the final accuracy by a non-trivial margin. This benefit is at least partly due to the model regularization effect of $\tau$ which alleviates over-fitting.

**Dataset.** We conduct a series of experiments on the task of image classification using the ImageNet-1K dataset (Deng *et al.*, 2009), which consists of 1.28 million training images across 1000 classes and 50k images for validation.

**Implementation details.** There are many variants of residual-like models used in the image classification task. We conduct experiments on three representative types of models, i.e. ResNet (He *et al.*, 2016), ViT (Dosovitskiy *et al.*, 2020) (and its variants, e.g., DeIT (Touvron *et al.*, 2021), Swin (Liu *et al.*, 2021)). During experiments, we find that all these models can learn an efficient Markov chain or even $\delta$-convex chain, so that only a very small $\tau$ (less than $10^{-6}$) is applied. All the models are trained for a 10-epoch linear warmup and a cosine decaying schedule afterward for 290 epochs. *More details can be found in the supplementary materials.*

**Result analysis.** As listed in Tab. 2, despite that the baseline models have already achieved a saturated accuracy, a suitable $\tau$ with penal connection routine can further push forward this performance

| # layers | 18 | 20 | 32 | 34 | 44 | 50 | 56 | 101 | 152 |
|---|---|---|---|---|---|---|---|---|---|
| CIFAR10 | | | | | | | | | |
| ResNet | - | 92.1 | 92.6 | - | 93.2 | - | 92.2 | - | - |
| Markov. | - | 92.3 | 93.2 | - | 93.4 | - | 93.4 | - | - |
| CIFAR100 | | | | | | | | | |
| ResNet | 76.3 | - | - | 77.6 | - | 78.5 | - | - | - |
| Markov. | 76.7 | - | - | 78.0 | - | 79.1 | - | - | - |
| ImageNet1k | | | | | | | | | |
| ResNet | - | - | - | - | - | - | - | 80.7 | 81.1 |
| Markov. | - | - | - | - | - | - | - | 81.1 | 81.4 |
| **Improv.** | **+0.4** | **+0.2** | **+0.6** | **+0.4** | **+0.2** | **+0.6** | **+1.2** | **+0.4** | **+0.3** |

Table 3: The accuracy of ResNet over different depths on CIFAR10, CIFAR100 and ImageNet1K.

by a non-trivial margin. Across our experiments, a large $\tau$ usually does hurt the mode performance. This observation reflects another effect of penal connection, that is $\tau$ not only engages the model to learn an efficient Markov chain, but it also limits the Markov chain to be too efficient, keeping the Markov chain away from a $\delta$-convex chain. We think it is also reasonable because a $\delta$-convex chain may lead to hard optimization, and a slight $\tau$ can take effect in this situation. In other words, the penal connection can also be viewed as a model regularization that can improve the final performance via alleviating over-fitting from a new standpoint.

### 3.3 MODEL DEGRADATION

One of the main reasons that residual-like models become widely used across various tasks is their capacity to counter the problem of model degradation in deeper networks. Model degradation refers to the phenomenon that with the increasing depth of model $\mathcal{M}_L$, the performance will no longer be improved and even worse. We find that by converting a residual-like model $\mathcal{M}_L$ to a Markov chain $\mathcal{C}_L$, the model degradation problem can be solved better. With the aid of the penal connection routine, a deep model can arrive at a higher performance than the original counterpart.

**Dataset.** CIFAR10 [2] dataset consists of 60k 32x32 color images in 10 classes. There are 50k training images and 10k test images. The CIFAR100 dataset is identical to CIFAR-10, except the number of classes is 100.

**Implementation details.** We take ResNet He *et al.* (2016) for comparison. In order to investigate the performance trend of models over different depths, we build seven models with different depths, i.e., $L \in \{18, 20, 32, 34, 44, 50, 56\}$. We used a momentum SGD optimizer with a batch size of 128 and a weight decay of 0.0001 for CIFAR10 and 0.0005 for CIFAR100. The $\tau$ is $3 \times 10^{-9}$ for all experiments. We trained all the models for 200 epochs from an initial learning rate of 0.1. The learning rate decayed by a factor of 10 at epochs 100, and 150 for CIFAR10, and at epochs 60, 120, and 160 for CIFAR100.

**Result analysis.** The results are listed in Tab. 3. All the Markov chain models significantly advance baseline residual models. In particular, as the model goes deeper, the performance of baseline models saturates first and decay afterward. On the contrary, the Markov chain with penal connection consistently achieves stable gains. This evidence confirms that our proposed method can further alleviate the model degradation problem, motivating future scaling efforts in depth.

## 4 CONCLUSIONS

In this work, we introduce the conception of a learnable Markov chain for the residual-like model, and propose a simple routine of penal connection to boost the model performance and alleviate the model degradation in deep depth as well. Adequate theoretical analysis and comprehensive experiments on the different types of architectures across a spectrum of tasks jointly demonstrate the rationality and effectiveness of the learnable Markov chain. While these initial results are encouraging, many challenges remain. For example, a better way to compute the ideal direction of Markov chain would likely lead to further improved performance. We expect more research would pay more attention to this new perspective which will inspire future work.

---

[2] https://www.cs.toronto.edu/~kriz/cifar.html

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

## A    RELATED WORKS

**The model with skip connection.**    In recent years, a large number of skip connection mechanisms have been used for neural network design (Liu *et al.*, 2020). Highway network (Zilly *et al.*, 2017) built a skip connection with a learnable gating mechanism from the input to the output. He *et al.* (2016) proposed the identity skip connections in ResNet, which have already become an indispensable building module in modern architecture design. This type of model with skip connections can be summarized as *a residual-like* model. In the field of natural language processing, *Transformer* (Vaswani *et al.*, 2017) made use of skip connection to build the multi-head self-attention module and the feed-forward network. Recently, Transformer has attracted extensive attention in computer vision and various variants (Han *et al.*, 2022) have been proposed which achieve superior performance against traditional CNNs. In this work, we aim to improve the performance of most existing residual-like models from the perspective of a learnable Markov chain.

**Markov chain.** A Markov chain or Markov process is a stochastic model describing a sequence of possible events in which the probability of each event depends only on the state attained in the previous event (Gagniuc, 2017). Informally, it can be thought of as "What happens next depends only on the state of affairs now.". Markov chains have many applications for statistical models (Karlin, 2014), such as studying cruise control systems in motor vehicles, queues or lines of customers arriving at an airport, currency exchange rates, and animal population dynamics (Meyn and Tweedie, 2009). Markov processes are the basis for general stochastic simulation methods known as Markov chain Monte Carlo, which are used for simulating sampling from complex probability distributions, and have found application in Bayesian statistics, thermodynamics, statistical mechanics, physics, chemistry, economics, finance, signal processing, information theory and speech processing (Gamerman and Lopes, 2006). Recently, the notation of the Markov chain also plays an important role in reinforcement learning (Otterlo and Wiering, 2012), image generation (Ho *et al.*, 2020) and other deep learning-related fields (Mardt *et al.*, 2018). Shwartz-Ziv and Tishby (2017) regards the feedforward process of a neural network as a Markov chain and explains the behavior of the neural network in the optimization process from the perspective of Shannon Information Theory.

## B PROOF OF LEMMA. 3

**Proof B.1** *According to the forward pass, we have*

$$\mathbb{E}[\|x_{l+1} - y\|_F^2] = \mathbb{E}[\|x_l - y + z_{l+1}\|_F^2] \tag{13}$$
$$= \mathbb{E}[\|x_l - y\|_F^2] + 2\langle x_l - y, d_{l+1}\rangle + \|z_{l+1}\|_F^2$$
$$\leq \mathbb{E}[\|x_l - y\|_F^2] + 2\langle x_l - y, d_{l+1}\rangle + Z^2$$
$$\leq (1 + 2\delta)\mathbb{E}[\|x_l - y\|_F^2] + Z^2. \tag{14}$$

*Now if $\delta\mathbb{E}[\|x_l - y\|_F^2] \geq Z^2$,[3] we know the $\mathbb{E}[\|x_l - y\|_F^2]$ will decrease by a factor of $(1 + \delta)$ for every chain node. Otherwise, although it could increase, we know*

$$\mathbb{E}[\|x_l - y\|_F^2] \leq \frac{Z^2}{\delta}. \tag{15}$$

*We know after $L$ nodes, either $\mathbb{E}[\|x_L - y\|_F^2]$ is already smaller than $\frac{Z^2}{\delta} = \frac{(1+a)\log L Z^2}{\delta^2 L}$, or it is decreasing by factor of $(1 + \delta)$ for every node, which means*

$$\mathbb{E}[\|x_L - y\|_F^2] \leq \mathbb{E}[\|x_0 - y\|_F^2](1 + \delta)^L$$
$$\leq D^2 e^{\delta L} = D^2 e^{(1+a)\log L} = \frac{D^2 L^a}{L}$$
$$\leq \frac{(1+a)\log L Z^2}{\delta^2 L}. \tag{16}$$

*The last inequality holds since*

$$L^a \log L \geq \frac{D^2 \delta^2}{(1+a)Z^2} \tag{17}$$

*Thus, $\mathbb{E}[\|x_L - y\|_F^2]$ will be smaller than $\frac{(1+a)\log L Z^2}{\delta^2 L}$.*

## C ANALYSIS OF ESTIMATION $g_{z_l}$

Here, we give a detailed analysis of the estimation error of $g_{z_l}$ in Eq. 12.

Firstly, we strictly follow the chain rule to derive the accurate formula of $g_{z_l}$:

---

[3]In order to simplify the derivation process, we use the F-norm function to measure the distance between each state $x_l$ and the target $y$ without losing the generality.

$$
\begin{aligned}
g_{z_l} &:= \frac{\partial \mathcal{L}_{\mathcal{M}_L}(x_L, y)}{\partial x_l} \frac{\partial x_l}{z_l} + \frac{\partial \lambda \|\epsilon\|_2^2}{\partial z_l} \\
&= g_{x_l} \frac{\partial(x_{l-1} + z_l)}{\partial z_l} + \lambda \frac{\partial \|\sum_{l=1}^{L} 1 + \eta \langle \vec{z}_l, -\vec{g}_{x_l} \rangle \|_2^2}{\partial z_l} \\
&= g_{x_l} + \lambda \frac{\partial(\|1 + \eta \langle \vec{z}_l, -\vec{g}_{x_l} \rangle \|_2^2)}{\partial z_l} \\
&= g_{x_l} + \lambda \frac{\partial(\|1 + \eta \langle \frac{z_l}{\|z_l\|_2}, -\vec{g}_{x_l} \rangle \|_2^2)}{\partial z_l} \\
&= g_{x_l} + 2\lambda(1 + \langle \frac{z_l}{\|z_l\|_2}, -\eta \vec{g}_{x_l} \rangle) \underbrace{\frac{\partial \langle \frac{z_l}{\|z_l\|_2}, -\eta \vec{g}_{x_l} \rangle}{\partial z_l}}_{:=A}
\end{aligned}
\tag{18}
$$

where

$$
\begin{aligned}
A &:= \frac{\partial \langle \frac{z_l}{\|z_l\|_2}, -\eta \vec{g}_{x_l} \rangle}{\partial z_l} \\
&= \frac{\partial \frac{\langle z_l, -\eta \vec{g}_{x+l} \rangle}{\|z_l\|_2}}{\partial z_l} \\
&= \frac{(-\eta \vec{g}_{x_l})\|z_l\|_2 + \|z_l\|_2^{-1} z_l \langle z_l, -\eta \vec{g}_{x_l} \rangle}{\|z_l\|_2^2} \\
&= -c\eta \vec{g}_{x_l} + c^3 \langle z_l, -\eta \vec{g}_{x_l} \rangle z_l.
\end{aligned}
\tag{19}
$$

$c := \frac{1}{\|z_l\|_2}$ is defined in Eq. 11. Then we substitute Eq. 19 into Eq. 18:

$$
\begin{aligned}
g_{z_l} &:= g_{x_l} + 2\lambda(1 + \langle \frac{z_l}{\|z_l\|_2}, -\eta \vec{g}_{x_l} \rangle)(-c\eta \vec{g}_{x_l} + c^3 \langle z_l, -\eta \vec{g}_{x_l} \rangle z_l) \\
&= g_{x_l} + 2\lambda(1 + \langle c z_l, -\eta \vec{g}_{x_l} \rangle)(-c\eta \vec{g}_{x_l} + c^3 \langle z_l, -\eta \vec{g}_{x_l} \rangle z_l) \\
&= g_{x_l} + 2\lambda(1 - c\eta t)(-c\eta \vec{g}_{x_l} - c^3 \eta t z_l)
\end{aligned}
\tag{20}
$$

where $t := \langle z_l, \vec{g}_{x_l} \rangle$. Eq. 20 can be further reformulated as:

$$
\begin{aligned}
g_{z_l} &:= g_{x_l} + 2\lambda(1 - c\eta t)(-c\eta \vec{g}_{x_l} - c^3 \eta t z_l) \\
&= (1 - 2c\eta \lambda \frac{1 - c\eta t}{\|g_{x_l}\|_2}) g_{x_l} + 2\eta \lambda c^3 t(c\eta t - 1) z_l
\end{aligned}
\tag{21}
$$

Hence, the estimated error term $\epsilon$ can be calculated by subtracting Eq. 12 from Eq. 21, which is:

$$
\epsilon := -2c\eta \lambda \frac{1 - c\eta t}{\|g_{x_l}\|_2} g_{x_l} + (2\eta \lambda c^3 t(c\eta t - 1) - \tau) z_l.
\tag{22}
$$

Since $\lambda$ and $\eta$ are very close to zero, the influence of the first term in right can be ignored, so the second term mainly contributes to the estimation error. When we choose a suitable hyper-parameter $\tau$ that makes $\tau = 2\eta \lambda c^3 t(c\eta t - 1)$ hold, the estimation error $\epsilon$ could be abysmally close to zero. Empirically, the value of $\tau$ varies significantly across different tasks.

## D  THE QUALITY OF THE PROPOSED IDEAL DIRECTION

There are many ways to define an ideal direction. As shown in Fig. 5, our proposed ideal direction calculation approach will satisfy the definition in most situations.

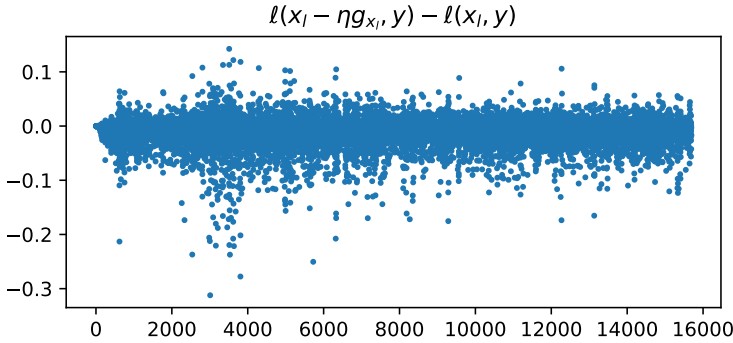

Figure 5: The difference between $\ell(x_l - \eta g_{x_l}, y) - \ell(x_l, y)$ during training in the toy model. Under more than $70.0\%$ of situations, $\ell(x_l - \eta g_{x_l}, y) < \ell(x_l, y$ strictly holds.

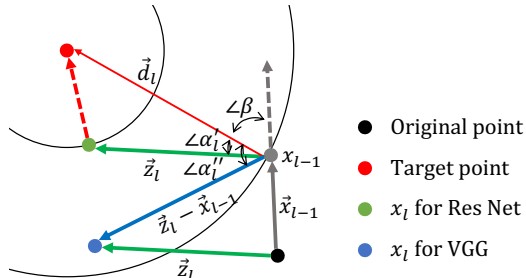

Figure 6: Advantage of ResNet over VGG under Markov chain's guide. As long as $\mathcal{C}_L(x_0 \to x_L)$ is a $\delta$-convex chain, the chain formed by ResNet is more efficient than the one formed by VGG.

## E  ADVANTAGE OF RESNET OVER VGG UNDER MARKOV CHAIN'S GUIDE

**Lemma 4** *If $\mathcal{C}_L(x_0 \to x_L)$ is $\delta$-convex, the chain $\mathcal{C}'$ formed by ResNet is more efficient than the chain $\mathcal{C}''$ formed by VGG, formally:*

$$\epsilon' \geq \epsilon'' \tag{23}$$

*where $\epsilon'$ is the efficiency of ResNet chain and $\epsilon''$ is the efficiency of VGG chain.*

**Proof E.1** *Without losing generality, we will discuss only one node here. As shown in Fig. 6, since $\mathcal{C}$ is $\delta$-convex, $x_l$ will always get closer to the target compared with $x_{l-1}$ and the original point. Hence,*

$$\langle \vec{x}_{l-1}, \vec{d_l} \rangle \geq 0 \tag{24}$$

*holds. Suppose the parameters $\theta_l$ are the same for ResNet and VGG, then:*

$$x'_l = z_l + x_{l-1} \tag{25}$$
$$x''_l = z_l, \tag{26}$$

*where $z_l := f_{\theta_l}(x_{l01})$. $x'_l$ and $x''_l$ are the next chain node for ResNet and VGG respectively.*

*As for chain node $x_{l-1}$, the estimated direction for ResNet is $\vec{z_l}$ and the estimated direction for VGG is $\vec{z_l} - \vec{x}_{l-1}$. Therefore,*

$$\epsilon' = \langle z_l, d_l \rangle \tag{27}$$
$$\epsilon'' = \langle z_l - x_{l-1}, d_l \rangle. \tag{28}$$

*According to Eq. 24, we have:*

$$
\begin{aligned}
&\langle x_{l-1}, d_l \rangle \geq 0 \\
\Rightarrow\ &\langle z_l, d_l \rangle - \langle z_l, d_l \rangle + \langle x_{l-1}, d_l \rangle \geq 0 \\
\Rightarrow\ &\langle z_l, d_l \rangle - \langle z_l - x_{l-1}, d_l \rangle \geq 0 \\
\Rightarrow\ &\langle z_l, d_l \rangle \geq \langle z_l - x_{l-1}, d_l \rangle \\
\Rightarrow\ &\epsilon' \geq \epsilon''
\end{aligned}
\tag{29}
$$

*Proofed.*

Lemma. 4 indicates that ResNet can form a more efficient Markov chain compared with VGG, and leads to better performance.

## F  MORE DISCUSSION ON MODEL DEGRADATION

As shown in Fig. 7, the randomly initialized Markov chain is prone to move in zigzags and even turn back as the chain goes longer (i.e. the model going deeper), which hinders the model from fitting target distribution efficiently, probably resulting in worse performance. When we apply penal connection to enforce the network to be an efficient Markov chain, the turn-back chain nodes do not exist so each node takes at least a non-negative effect whatever how long the chain is. As a result, it could alleviate the model degradation problem.

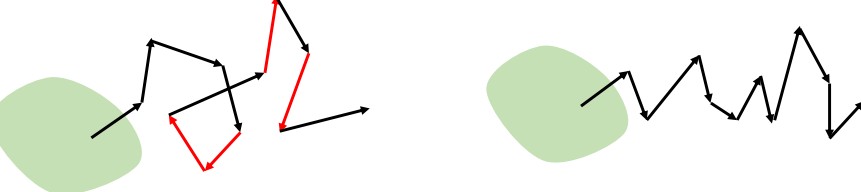

(a) Folding Markov chain.          (b) Unfolding Markov chain.

Figure 7: Structural comparison between a folding Markov chain (a) and an unfolding Markov chain (b).

