# OpenReview forum: "Rethinking skip connection model as a learnable Markov chain"
_ICLR.cc/2023/Conference — ICLR 2023 poster_

### Official Review · Reviewer_Pv9F · 2022-10-22

**Confidence:** 4
**Correctness:** 3
**Technical Novelty And Significance:** 3
**Empirical Novelty And Significance:** 3
**Recommendation:** 6

**Clarity, Quality, Novelty And Reproducibility:**

[1] This paper is clearly written and well organized.

[2] The main claim is well supported by either proof or experimental results.

[3] Rethinking the skip connection model as a learnable Markov chain is novel.

[4] The method is easy to reproducible.


**Strength And Weaknesses:**

Strength:

[1] Interpret the skip connection with the help of the Markov chain is reasonable and innovative.

[2] When exploring the optimization of the Markov chain, the gradient to $z_l$ seems to add a simple penalization on the $z_l$, which is interesting and easy to use.

[3] This paper evaluated their method on both Natural Language Processing and Computer Vision. Results show that their method improves the performance and the convergence of deep neural networks.

Weaknesses:

[1] Can authors analyze why the proposed method can solve the model degradation problem? Will the penal connection make the convolutional layer more sparsity, and can other regularization methods (i.e. $L_1$ Norm) solve this problem?

[2] Experiments on more deeper networks (i.e. ResNet101, ResNet152, ResNet110, ResNet1202) are needed to evaluate for demonstrating that the Markov chain can better solve the model degradation problem than the residual-like model.

[3] I have noticed ReLU is put before Linearly Layer in Figure 1 in the supplementary materials, so the model used in the main paper is whether ResNets or PreActResNets?

**Summary Of The Paper:**

This paper indicates that a model with residual-like bocks can also be considered as a learnable Markov chain.
They prove that a Markov chain can shift source input to the target domain through L nodes in an efficient way if the Markov chain is $\delta$-convex.
By solving a Markov chain optimization problem they introduce the panel connection method, which is acting like a model regularization.
Finally, they conduct experiments to demonstrate the superiority of the Markov chain.

**Summary Of The Review:**

This paper rethinks the skip connection model as a learnable Markov chain and proposes a novel penal connection for better optimization.
Most of the theorem and experiments are satisfied. However, there are still some concerns as written in "Strength And Weaknesses".

---

> ### Author Response · Authors · 2022-11-19
> **Q1: Can authors analyze why the proposed method can solve the model degradation problem?**
>
> We explain this phenomenon in Fig. 7 of Appendix F in the revision. From the illustration, the random initialized Markov chain is prone to move in zigzags and even turn back as the chain goes longer (i.e. the model going deeper), which hinders the model from fitting target distribution efficiently, probably resulting in worse performance.
>
> When we apply penal connection to enforce the network to be an efficient Markov chain, the turn-back chain nodes do not exist so each node takes at least a non-negative effect whatever how long the chain is. As a result, it could alleviate the model degradation problem.

---

> ### Author Response · Authors · 2022-11-19
> **Q2: Will the penal connection make the convolutional layer more sparsity, and can other regularization methods (i.e.  Norm) solve this problem?**
>
> Penal connection can be viewed as a special type of regularization acted on gradients of featuremaps, instead of parameters. Thus, it can not directly affect the sparsity of weights.
>
> Moreover, the penal connection is a simple approach to guide a network to be a more efficient Markov chain. We believe that there must be other solutions to do better, which is worth exploring in future work.

---

> ### Author Response · Authors · 2022-11-19
> **Q3: Experiments on deeper networks (i.e. ResNet101, ResNet152, ResNet110, ResNet1202) are needed to evaluate for demonstrating that the Markov chain can better solve the model degradation problem than the residual-like model.**
>
> Here, we conduct experiments on ImageNet1K with ResNet101 and ResNet152. We can observe that models with the penal connection can consistently improve performance as the depth goes deeper. (The penal connection $\tau$ set as $3\times 10^{-8}$.)
>
> | Model     | baseline  | ours  |
> | :--:      | :--:      | :--:  |
> | ResNet101 |  80.7     | 81.1 |
> | ResNet152 |  81.1     | 81.4 |

---

> ### Author Response · Authors · 2022-11-19
> **Q4: I have noticed ReLU is put before Linearly Layer in Figure 1 in the supplementary materials, so the model used in the main paper is whether ResNets or PreActResNets?**
>
> Probably there is a misunderstanding. Fig. 1 in the supplementary materials demonstrates the network structure used in the toy model mentioned in Figure 2(b) of the main paper, which is not a model related to ResNet. We only use ResNet anywhere.

---

### Official Review · Reviewer_CKAs · 2022-10-24

**Confidence:** 3
**Clarity, Quality, Novelty And Reproducibility:** The paper is well writing and easy to…
**Correctness:** 3
**Technical Novelty And Significance:** 3
**Empirical Novelty And Significance:** 3
**Recommendation:** 6

**Strength And Weaknesses:**

Pros

- This paper formulated models with skip connections as a learnable Markov chain. And introduced the ideal direction for improving the efficiency of the Markov chain.
- The proposed penal connection is a plug-in operation and the experiment evaluation the efficiency of this method.

Cons

- Previous works [1,2] have already formulated the VGG-like models with the Markov chain, can you explain the advantage of ResNet over VGG under the Markov chain’s guide?
- The compared models’ performance in Table 2 is lower than their original paper, can you explain why, and can the proposed method promote their SOTA implementation?

[1] Opening the black box of Deep Neural Networks via Information

[2] Markov Chain Neural Networks

**Summary Of The Paper:**

This paper rethinks the skip connection model as a learnable Markov chain and proposes a penal connection to convert a residual-like model to a Markov chain for more efficient training. Experiments on MLP and CV demonstrate the superiority of this plug-in method.

**Summary Of The Review:**

- This paper missing the comparison with the previous close-related methods.
- This paper needs to compare their performance on a better-implied baseline.

---

> ### Author Response · Authors · 2022-11-19
> **Q1: Previous works [1,2] have already formulated the VGG-like models with the Markov chain.  [1] Opening the black box of Deep Neural Networks via Information; [2] Markov Chain Neural Networks**
>
> First, [2] proposes a better way of Markov process simulation with a neural network. This is a new application of neural networks. Our work mainly discusses the optimization behavior of neural networks. They belong to two distinct directions and have no obvious correlation.
> Second, [1] explains the learning behavior of neural networks from different perspectives. Specifically, [1] regards the feed-forward process of a neural network as a Markov chain as well, and then explains the behavior of the neural network in the optimization process from the perspective of Shannon Information Theory. In comparison to our work,
> we explain the behavior in the process of network training from the perspective of an effective Markov chain.
> And we subsequently propose the resultant penal connection that improves performance by explicitly guiding any residual-like network within a more efficient Markov chain.

---

> ### Author Response · Authors · 2022-11-19
> **Q2: Can you explain the advantage of ResNet over VGG under the Markov chain’s guide?**
>
> This is a very good question and should be fully discussed. A detailed analysis is exhibited in Appendix E in the revision.
> Briefly, as long as $\mathcal{C}_{L}(x_{0} \to x_{L})$ is a $\delta$-convex chain, the chain formed by ResNet is more efficient than the one formed by VGG. This is the main reason why ResNet is superior to VGG with respect to the Markov chain.

---

> ### Author Response · Authors · 2022-11-19
> **Q3: The compared models’ performance in Table 2 is lower than their original paper, can you explain why, and can the proposed method promote their SOTA implementation?**
>
> In Experiment 2, we mainly focused on the impact of different $\tau$ on the model performance.
> Therefore, we simply take the same training recipes for all networks, except for different batch sizes and corresponding learning rates, which is not suitable for these networks to obtain superior accuracy, resulting in worse performance than presented in their original paper.
>
> We have rerun the experiments with their optimal training recipes to catch up with the SOTA performance.
> Due to the limitation of rebuttal period time and computation resources, only part of the results has come out at present. We observe penal connection can still improve the SOTA performance. (The penal connection $\tau$ equals to $3\times 10^{-8}$ for all experiments.)
>
> | model         | baseline  | ours  |
> | :--:          | :--:      | :--:  |
> | ResNet50      |  79.2     | 79.4 |
> | ViT-S/16-224  |  77.8     | 78.1 |
> | Swin-S/4-7-224|  83.3     | 83.5 |

---

### Official Review · Reviewer_Z5HW · 2022-10-24

**Confidence:** 4
**Correctness:** 3
**Technical Novelty And Significance:** 3
**Empirical Novelty And Significance:** 3
**Recommendation:** 6

**Clarity, Quality, Novelty And Reproducibility:**

The paper is well organized and presents a new perspective on residual networks. I think it's great research.

**Strength And Weaknesses:**

Strengths:
This paper has a solid theoretical foundation and novel method.
Weaknesses:
1、	The main contribution of this paper has not been summarized.
2、	In Fig.1, f_{\theta_{l}} is a residual-like block. Can you explain the difference between f_{\theta_{l} and residual block?
3、	How to understander the definition of ideal direction? The relationship between ℓ(a, c) ≥ ℓ(µa+ (1−µ)b, c) ≥ ℓ(b, c) and Eq.(1).
4、	In Fig.4, can you plot the curves about ℓ(xl − ηgxl , y) and  ℓ(xl , y)  to illustrate ℓ(xl − ηgxl , y) < ℓ(xl , y) always holds?

**Summary Of The Paper:**

This paper reformed skip connections into a learnable Markov chain and proposed a simple routine of penal connection to make any residual-like model become a learnable Markov chain. The ablation analysis and the compared experiments showed that the proposed method achieves more competitive results than the SOTA methods. Overall, this paper has a solid theoretical foundation and novel method.

**Summary Of The Review:**

This paper reformed skip connections into a learnable Markov chain and proposed a simple routine of penal connection to make any residual-like model become a learnable Markov chain. The ablation analysis and the compared experiments showed that the proposed method achieves more competitive results than the SOTA methods. Overall, this paper has a solid theoretical foundation and novel method. I think it can be accepted.

---

> ### Author Response · Authors · 2022-11-19
> **Q1: The main contribution of this paper has not been summarized.**
>
> Our main contributions can be summarized in two folds.
> First, we present a new perspective to understand the skip connection model as a learnable Markov chain and carry out exhaustive theory analysis and experimental verification.
> Second, we propose the penal connection, a simple method to enable a network to be optimized within a more efficient Markov chain, which can substantially improve performances both in the fields of NLP and CV.
> (See also in the last paragraph of Section: Introduction in the revision.)

---

> ### Author Response · Authors · 2022-11-19
> **Q2: In Fig.1, $f_{\theta_{l}}$ is a residual-like block. Can you explain the difference between $f_{\theta_{l}}$ and residual block?**
>
> Generally speaking, the residual block refers to the basic block and bottleneck block proposed in the original ResNet paper.
> A residual-like block is a broader concept that refers to the block that several computational operations embraced with a skip connection, such as the self-attention block and FFN block in Transformer.

---

> ### Author Response · Authors · 2022-11-19
> **Q3: How to understand the definition of ideal direction? The relationship between ℓ(a, c) ≥ ℓ(µa+ (1−µ)b, c) ≥ ℓ(b, c) and Eq.(1).**
>
> Given the error function ℓ, relative to direction $\vec{a}$, any direction $\vec{b}$, which satisfies that ℓ(b) is less than ℓ(a), is the ideal direction of $\vec{a}$.
> In other words, whether $\vec{b}$ is an ideal direction of $\vec{a}$ depends on the definition of error function ℓ.
> Actually, direction $\vec{a}$ and $\vec{b}$ lie in the same linear space, and the interpolation between $\vec{a}$ and $\vec{b}$ also lies in this space and `ℓ(a, c) ≥ ℓ(µa+ (1−µ)b, c) ≥ ℓ(b, c)` reflects this property.

---

> ### Author Response · Authors · 2022-11-19
> **Q4: In Fig.4, can you plot the curves about ℓ(xl − ηgxl , y) and ℓ(xl , y) to illustrate ℓ(xl − ηgxl , y) < ℓ(xl , y) always holds?**
>
> Since the parameters of the network are constantly changing during the training process, it is almost impossible to require that `ℓ(xl − ηgxl , y) < ℓ(xl , y)` always be established.
> The goal of penal connection is not to ensure `ℓ(xl − ηgxl , y) < ℓ(xl , y)` always holds, but to obtain a suitable efficient Markov chain.
> In order to better illustrate this relationship, we plot the curves about `ℓ(xl − ηgxl , y) - ℓ(xl , y)` in Appendix D of the revision for the toy model mentioned in Fig. 2 (b).
> In most cases, `ℓ(xl − ηgxl , y)` is significantly smaller than `ℓ(xl , y)` and in case of more than 70\%, `ℓ(xl − ηgxl , y) < ℓ(xl , y)` holds.
> This fully demonstrates the effectiveness of our proposed penal connection.
> However, finding a better way to make `ℓ(xl − ηgxl , y) < ℓ(xl , y)` hold is worth further exploration.

---

### Official Review · Reviewer_Geje · 2022-10-30

**Confidence:** 3
**Correctness:** 3
**Technical Novelty And Significance:** 3
**Empirical Novelty And Significance:** 3
**Recommendation:** 6

**Clarity, Quality, Novelty And Reproducibility:**

Quality: The paper is interesting, however some central claims are not clear (see above).

Clarity: Clarity can be improved, especially with regards to the central claims of the paper.

Originality: To the best of my knowledge, the proposed idea is novel.


**Strength And Weaknesses:**

Strengths:
* The Markov Chain perspective on skip connection type models is novel and interesting.
* The paper includes useful illustrations e.g. Fig 2 and Fig 3, which make the paper clearer.
* The paper includes encouraging results on top-1 accuracy on ImageNet-1k across several state of the art models.

Weaknesses:
* Central claims of the paper are not clear: Consider the “inefficient” Markov chain in Fig 2 (a). Both the inefficient and efficient Markov chains successfully convert data from the source to the target domain. In fact, for a classification type model there should be no difference in the finally test accuracies of both efficient and inefficient models. If learning an efficient Markov chain is hard, it is not clear what is the advantage of an efficient Markov chain. Perhaps the paper wants to claim that efficient Markov chains leads to better fit to the target distribution?
* In Proof 2.3, the paper derives only an c value for \epsilon. What is the quality of the approximation? Are there any upper bounds to the absolute error? This is important because the approximation has a direct impact on the quality of the proposed regularizer. (The steps in Proof 2.3 should be numbered).
* It is not clear if a higher value of \epsilon leads to a better model? The paper should show better evidence for this. In Fig 4 (b) the difference in \epsilon between the two models is minimal. In fact, the value of \epsilon peaks early in training, but is not accompanied by peak accuracy?
* The paper should also consider including additional state of the art baselines for the machine translation task, as currently only the plain Transformer is considered.

**Summary Of The Paper:**

This paper formulates skip connection based models a learnable Markov chain. The paper then introduces the content of an efficient Markov chain which maps data from the input to the target domain efficeintly. A simple regularisation scheme is introduce to enable the learning of efficient Markov chains. Evaluation is performed on Transformer and ResNet type models.

**Summary Of The Review:**

The paper is interesting, however it is not clear if higher \epsilon always leads to better test accuracy and whether having a higher \epsilon is actually necessary. Moreover, the quality of the approximation of \epsilon considered in the paper should be quantised with an upper bound of the error.

---

> ### Author Response · Authors · 2022-11-19
> **Q1: Central claims of the paper are not clear: Consider the “inefficient” Markov chain in Fig 2 (a). Both the inefficient and efficient Markov chains successfully convert data from the source to the target domain. In fact, for a classification-type model, there should be no difference in the final test accuracies of both efficient and inefficient models. If learning an efficient Markov chain is hard, it is not clear what is the advantage of an efficient Markov chain.**
>
> Thank you for pointing this out. There is a mistake in the illustration of Fig 2(a) that makes you confused. We claim that efficient Markov chains can better fit the target distribution, leading to superior accuracy. We have corrected the figure in the revision.
>
> Because most existing optimizers are inadequate to keep a network in an efficient Markov chain, the network generally shows substandard optimizability.
> Therefore, if there is a way, i.e., the proposed penal connection, to explicitly guide the network to be a more efficient chain in the optimization process, it will probably boost the performance.

---

> ### Author Response · Authors · 2022-11-19
> **Q2: In Proof 2.3, the paper derives only a c value for $\epsilon$. What is the quality of the approximation? Are there any upper bounds to the absolute error? This is important because the approximation has a direct impact on the quality of the proposed regularizer. (The steps in Proof 2.3 should be numbered).**
>
>  It is an excellent suggestion to dive more profound into the approximation of $\epsilon$. A detailed analysis of this approximation is presented in Appendix C in the revision.
> We proved that the estimation error of $g_{x_{l}}$ is mainly contributed by the hyper-parameter $\tau$.
>
> All equations are correctly numbered now.

---

> ### Author Response · Authors · 2022-11-19
> **Q3: It is not clear if a higher value of $\epsilon$ leads to a better model. The paper should show better evidence for this.**
>
> The value of $\epsilon$ only represents how efficient a Markov chain is, it is not directly corresponding to model accuracy. Consequently, a higher value of $\epsilon$ does not mean higher accuracy. Thus, the value of $\epsilon$ peaks early in training also does not mean that the model will perform optimally at this time.

---

> ### Author Response · Authors · 2022-11-19
> **Q4:In Fig 4 (b) the difference in $\epsilon$ between the two models is minimal. In fact, the value of $\epsilon$ peaks early in training, but is not accompanied by peak accuracy.**
>
> Although a more efficient Markov chain ideally leads to better performance, it increases the difficulty of optimization as well. As shown in Fig.4(b), the $\epsilon$ grows rapidly at the beginning of training, which implies that the penal connection mechanism successfully forces the network to be an extremely efficient chain. And then the $\epsilon$ falls sharply and keeps a small positive value close to zero, which is caused by the limitation of existing optimization algorithms that can not optimize the network with an efficient Markov chain. Despite this, the penal connection still helps models get better accuracy.

---

> ### Author Response · Authors · 2022-11-19
> **Q5: The paper should also consider including additional state-of-the-art baselines for the machine translation task, as currently only the plain Transformer is considered.**
>
> Our goal is to validate the effectiveness of the view of the effective Markov chain proposed in this work and improve the network performance by an explicit intervention that makes it to be an effective Markov chain. It is not our purpose to beat all existing methods on a certain task to achieve a new SOTA. Consequently, we conducted several representative tasks both in the fields of NLP and CV and examine the performance improvements of baseline models cooperated with the proposed penal connection.

---

### Author Response · Authors · 2022-11-19
**Thanks and a brief summary of responses**

Dear Reviewers,

Thank you for the valuable comments and constructive suggestions.

We are encouraged to see a wide appreciation of the novel view of the efficient Markov chain and penal connection. We have provided detailed responses to each reviewer, and `a revision with all review comments/suggestions addressed has been submitted`. By doing so, we hope that the reviewer's concerns could be largely addressed.

Thank you,

The authors.

---

### Decision · Program_Chairs · 2023-01-20

**Decision:**

Accept: poster

**Justification For Why Not Higher Score:**

Paper looks reasonable for acceptance, but not terribly strong. I personally doubt this work will be impactful but some people enjoy reading theoretical stuff so it should probably be a poster for folks who are interested.

**Justification For Why Not Lower Score:**

Reviewers said that the paper is novel and interesting. So i think it could be accepted.

**Metareview: Summary, Strengths And Weaknesses:**

This paper proposes a learnable markov chain view as an explanation to ResNets.

They conduct experiments on image classification and translation to verify that their proposed "view" works.

I think this is an okay paper that is interesting enough to be accepted, although I wouldn't be too upset if it was rejected.

Reviewers found the paper novel and said that  " This paper has a solid theoretical foundation and novel method."  and the main claim is supported....I have no clue tbh, so i am going to trust the reviewers on this one.

There seems to be some discussion going on which is healthy.

I hope this paper gets some impact in some way.

Anyways, i recommend a weak accept so good luck!

**Note From Pc:**

if the above contains the word "oral" or "spotlight" please see: "oral" presentation means -> notable-top-5% and "spotlight" means -> notable-top-25%. As stated in our emails, we are disassociating presentation type from AC recommendations